# Development of SLAF-Sequence and Multiplex SNaPshot Panels for Population Genetic Diversity Analysis and Construction of DNA Fingerprints for Sugarcane

**DOI:** 10.3390/genes13081477

**Published:** 2022-08-19

**Authors:** Hui Zhang, Pingping Lin, Yanming Liu, Chaohua Huang, Guoqiang Huang, Hongtao Jiang, Liangnian Xu, Muqing Zhang, Zuhu Deng, Xinwang Zhao

**Affiliations:** 1National Engineering Research Center for Sugarcane, Key Laboratory of Sugarcane Biology and Genetic, Breeding Ministry of Agriculture, Fujian Agriculture and Forestry University, Fuzhou 350002, China; 2Guangxi Key Laboratory for Sugarcane Biology, Guangxi University, Nanning 530005, China

**Keywords:** sugarcane, genetic diversity, DNA fingerprinting, variety identification, SNP

## Abstract

A genetic diversity analysis and identification of plant germplasms and varieties are important and necessary for plant breeding. Deoxyribonucleotide (DNA) fingerprints based on genomic molecular markers play an important role in accurate germplasm identification. In this study, Specific-Locus Amplified Fragment Sequencing (SLAF-seq) was conducted for a sugarcane population with 103 cultivated and wild accessions. In total, 105,325 genomic single nucleotide polymorphisms (SNPs) were called successfully to analyze population components and genetic diversity. The genetic diversity of the population was complex and clustered into two major subpopulations. A principal component analysis (PCA) showed that these accessions could not be completely classified based on geographical origin. After filtration, screening, and comparison, 192 uniformly-distributed SNP loci were selected for the 32 chromosomes of sugarcane. An SNP complex genotyping detection system was established using the SNaPshot typing method and used for the precise genotyping and identification of 180 sugarcane germplasm samples. According to the stability and polymorphism of the SNPs, 32 high-quality SNP markers were obtained and successfully used to construct the first SNP fingerprinting and quick response codes (QR codes) for sugarcane. The results provide new insights for genotyping, classifying, and identifying germplasm and resources for sugarcane breeding

## 1. Introduction

Sugarcane (*Saccharum* spp.), a C4 plant, is mainly distributed in subtropical and tropical areas [1]. An important sugar and energy crop, sugarcane is the source for 80% of the sugar and 40% of the ethanol worldwide [2]. The narrow genetic basis, blood grid, and inbreeding of sugarcane present challenges for germplasm identification and protection of different varieties [3]. For example, the same varieties may be referenced by different names at different breeding institutions, or different varieties may have the same name. Moreover, since the genetic relationship between varieties is not clear, identification based on morphology cannot be performed [4]. Molecular markers have been widely used in plant breeding [5]. Different types of markers have been successfully used with hybridization-based techniques involving restriction fragment length polymorphisms (RFLPs) [6], as well as PCR-based techniques using random amplification of polymorphic DNA (RAPD) [7], amplified fragment length polymorphism (AFLP) [8], and simple sequence repeats (SSRs) [9] for the evaluation of crop germplasms and to identify different varieties. Recently, sequencing-based high-throughput SNPs were shown to be valuable for genetic studies of different crops [10]. SNPs played an important role in large-scale population studies of wheat [11], rice [12], maize [13], and other crops [14]. SNPs are the foremost ample and universal sequence variations of genomes, which makes them helpful markers for genetic analyses in plants. SNP markers can be used to create plant DNA fingerprints since they are plentiful, broadly distributed, and have high stability and repeatability.

The advent of next-generation sequencing (NGS) technologies has caused the number of genetic variants that can be found in a single experiment to exponentially grow [15]. The sugarcane organization and QTLs associated with SCYLV resistance have been successfully built using an Axiom Sugarcane100K SNP array [16]. *Saccharum spontaneum* L.’s entire ordering sequence was published, hastening the finding of SNPs and genetic mapping and allowing for the further building of maps from short reads of different genotypes using the ordering sequence as a reference [17]. The SLAF-seq method, one of the SNP sequencing methods, has come into prominence as a replacement strategy to reduce restrictions associated with previously employed markers [18]. A powerful, superior, effective, and simple method for collecting thousands of markers from a large number of individuals is SLAF-seq [19]. It also allows SNPs to be identified using a small illustration library. SLAF-seq is typically utilized in plant breeding and is frequently used in mapping (linkage and association) research, genomic selection, and genetic diversity investigations.

Several common SNP genotyping methods based on electrophoresis, such as CAPS (cleaved amplified polymorphic sequence) [20], dCAPS (derived CAPS) [21], and AS-PCR (allele-specific PCR) [22], are commonly used in genetic research. However, some loci cannot be easily detected and genotyped using these SNP genotyping platforms since the regions flanking these SNPs are not conserved or there are other hits in the genome due to the sequence homology. This lack of conservation and sequence homology may lead to false positive or negative results in SNP genotyping. Thus, selecting corresponding markers according to different research objectives and the complexity of the crop genome is critical. Although the cost of whole-genome sequencing has fallen substantially during the last decade, complete sequencing of all genotypes for routine screening of breeding materials, especially for species—like sugarcane—that have a very large genome, is impractical for most breeding programs [23]. Gao et al. utilized 148,500 polymorphic SLAF-SNPs and SSR markers to construct a physical genetic map for sugarcane, and 77 SNPs from major quantitative trait locus (QTLs) were successfully transformed into 5 kompetitive allele-specific PCR (KASP) markers related to smut resistance. These KASP markers represent useful tools for marker-assisted sugarcane breeding [24]. However, the low success rate of primer design in sugarcane application is due to the need of sequences on both sides of the site for primers based on KASP labeling.

SNaPshot Multiplex technology, which was developed by the Applied Biology Corporation (ABI), is based on the fluorescence-labeling single base extension principle, also known as small sequencing. The Multiplex SNaPshot panels system, whilst slightly more expensive than other available typing techniques, such as KASP, has the advantages of good consistency, high throughput, and strong site flexibility. SNaPshot is an easily integrable SNP typing option for forensic laboratories that has readily available forensic human and non-human DNA assays [25]. At present, SNaPshot has been widely used in bacterial community identification [26], human disease correlation [27], and forensic medicine, but to date has not been widely used for plant genetics. In *Arabidopsis thaliana*, 17 uniformly-distributed SNP markers were constructed using multiple SNaPshot techniques were used to identify different varieties [28]. In melon, the expression sequences of the parental line were re-sequenced, and a physical genetic map with uniform distribution was developed using SNaPshot. This map laid a foundation for the later construction of a genetic map of this cucurbitaceous plant [29]. In tea plants, 10 SNPs were selected to transform by SNaPshot to genotype 20 tea germplasms. The results revealed good performance for the differentiation and identification of tea plants. Multiplex PCR and SNaPshot typing methods were used to develop a composite typing detection system for differentiation in other crops [30].

In sugarcane, many improved cultivars and modern varieties were developed by both public and private organizations [31]. However, distinguishing among the similar sugarcane varieties available in the trading market is difficult. The International Union for the Protection of New Varieties of Plants (UPOV) has established “distinctness, uniformity, and stability” (DUS) testing for new varieties before registration [32,33]. DNA fingerprinting constructed from molecular markers can provide better and more stable genetic information about different varieties [34]. In maize, a high-throughput and compatible SNP array was developed and successfully used to construct DNA fingerprints, resource evaluation, and molecular marker-assisted breeding [35]. In common bean (*Phaseolus vulgaris* L.), 754 markers were used to define clustering trees for 708 genotypes, and a fingerprint recognition platform was constructed [36]. In tobacco, 43 core markers were developed and used to construct SNP fingerprints for 216 cigar germplasm accessions [37].

In this study, SLAF-seq data from 103 sugarcane germplasm resources were used for genetic diversity analysis, and SNP sites that have good polymorphisms that were uniformly distributed across the entire genome were identified by filtering and screening. A total of 43 markers, including 32 core SNPs and 11 candidate SNPs, were successfully developed by SNaPshot technology and a fingerprint for 180 sugarcane varieties was constructed.

## 2. Materials and Methods

### 2.1. Plant Materials

A total of 103 sugarcane materials from Fujian Agriculture and Forestry University (Appendix A) were collected for SLAF sequencing. Of the 103 sugarcane materials, 79 represent germplasm from high-quality sugarcane-producing regions such as Fujian, Guangxi, Guangdong, Taiwan, and Yunnan provinces of China and overseas countries. In addition, 24 are fine-stem wild sugarcane materials (*Saccharum spontaneum* L.). A total of 180 sugarcane accessions were used for core SNP marker verification and DNA fingerprinting construction. These materials were comprised of breeding lines, landraces, commercial hybrids, and wild germplasm. They were provided by the Key Lab of Sugarcane Biology and Genetic Breeding (119°24′ E longitude, 26°08′ N latitude), Nanning Germplasm Resources Nursery of Guangxi University (108°25′ E longitude, 22°85′ N latitude), and Hainan Sugarcane Breeding Farm (109°17′ E longitude, 18°36′ N latitude) (Appendix A).

### 2.2. DNA Extraction and SLAF Library Construction and Sequencing

A total of 103 sugarcane samples were subjected to simplified genome sequencing using SLAF-sequence. For each germplasm, about 0.5 g of fresh, young, and veinless leaves were extracted with the cetyl ammonium bromide (CTAB) method [38]. NanoDrop2000 UV spectrophotometer (Thermo Fisher Scientific, Waltham, MA, USA) and 3%-agarose gel were used for DNA quantification.

We selected the reference genome of sugarcane for electronic digestion prediction [17]. DNA samples (100 ng/reaction) were double-digested with RsaI and EcoRⅤ-HF^®^ (New England Biolabs, Ipswich, MA, USA) and the sequence of fragment lengths 464–494 was defined as a SLAF-tag. The principles of the selective digestion plan are as follows: (1) The proportion of digestion SLAF tags located in repeat sequences should be as low as possible; (2) the SLAF tags were evenly distributed on the genome as far as possible; (3) the consistency of the fragment length should be maintained with the specific test system [39]; and (4) the number of enzyme-cut SLAF tags obtained met the expected number of tags. All qualified samples were digested by enzymes, respectively. The obtained enzyme SLAF tags were treated by adding one A at the 3‘ end, thereby connecting the dual index [40] sequencing connector, PCR amplification, purification, sample mixing, and gel cutting to select the target fragment. SLAF tags of 464–494 bp were gel-purified for sequencing. Pair-end sequencing was done using the Illumina high-throughput sequencing platform (Illumina, San Diego, CA, USA).

### 2.3. SLAF-Seq Data Grouping and Genotyping

Grouping and genotyping of SLAF-seq data were carried out by the procedures described by Sun et al. [18] Briefly, the high-quality reads (error chance < 0.01%, QC30) were mapped onto the sugarcane reference genome (*S. spontaneum* genome AP85-441, ftp://apple.fafu.edu.cn/SsponAnnotation, access on 4 May 2019) using BWA 0.7.10-r789 software [41,42]. The reads with over 90% identity were considered as identical SLAF markers and grouped in one SLAF locus. The Genome Analysis Toolkit 3.8 [43] and SAMtools 0.1.18 [44,45] were applied for SNPs identification. PLINK 1.07 [46] was used to filter high-quality SNPs for subsequent analysis. In addition, the filtered high-quality SNPs were used to annotate the SNP detection results with SnpEFF 4.3 [47] software.

### 2.4. Data Analysis

TASSEL software was used to calculate eigenvectors and eigenvalues according to SNP differences between individuals. PCA of the SNP data of 103 accessions was performed using the R 4.2.1 software package, SNPrelate [48]. A population structure analysis was performed using admixture 1.3.0. software [49] based on the maximum-likelihood method and *K* ranging from 1 to 10. The cross-validation error rate of the *K* value was analyzed. The phylogenic trees from the SNP data were constructed by the neighbor-joining method with a bootstrap value of 1000 using MEGA 7.0 software [50]. The visualization and annotation of the phylogenic trees were completed through the Evolview platform (https://www.evolgenius.info/evolview/#/treeview, accessed on 13 April 2022 ).

### 2.5. Development of SNaPshot Primers

Between 4 to 8 SNPs were evenly selected according to the length of each chromosome. The following steps were performed to convert it to SNP markers: (1) TBtools software was used to compare the sugarcane reference genome and select a 200-bp sequence above and below the SNP locus. (2) The BLAST function in NCBI was used to detect the homology of sequences. The sequence with multiple homology SLAF tags was eliminated. (3) Specific primers were designed according to the flanking sequence of the target SNPs to amplify the fragment containing the SNP locus. Exonuclease I (Exo I) and shrimp alkaline phosphatase (SAP) (TaKaRa, Japan) were added to the amplification product, and the primer sequences and the remaining dNTPs in the reaction system were digested. (4) Using the purified amplified product as a template, a unidirectional extension primer was designed according to the upstream or downstream sequence adjacent to the site. The first base at the 3‘ end of the primer was adjacent to the site to be tested, and the Tm value was above 50 °C. Poly C or Poly T of different lengths were added to the 5′ end of the primer.

SNPs were detected by ABI’s SNaPshot Multiplex Kit. The total reaction system was a 20-μL PCR reaction system, including 1 μL of genomic DNA (10 ng/μL) 1.2 μL of MgCl_2_ (25 mM), 2 μL10× of PCR Buffer, 2.4 μL of dNTP (2.5 mm), 1.0 μL of Probe Mix, 0.2 μL of HotStarTaq (5 U/μL), and 12.2 μL of ddH_2_O. The PCR reaction procedures were as follows: pre-denaturation at 95 °C for 2 min, denaturation at 94 °C for the 20 s, annealing at 65 °C for 40 s (−0.5 °C/cycle), extension for 90 s at 72 °C for 11 cycles; 94 °C denaturation for 20 s, 59 °C annealing for 30 s, 72 °C extension for 90 s for 24 cycles; and the last 72 °C extension for 2 min. PCR products were purified by reference to ABI’s SNaPshot technology. The extended PCR reaction procedure is as follows: pre-denaturation at 96 °C for 1 min, denaturation at 96 °C for 10 s, annealing at 52 °C for 5 s, and extension at 60 °C for the 30 s carried out for 28 cycles. Finally, ABI’s PRISM3730 sequencer was used for SNP genotyping and GeneMapper 4.1 [51] software was used for typing results statistics.

### 2.6. SNP Selection and Generation of QR Codes

Based on the SNP information obtained from the above analysis, 4-8 SNP loci were selected on each chromosome and 192 SNP markers were selected in total. Primers were developed by SNaPshot technology. Those primers with existing polymorphic differences were selected as core SNP markers, and primers with existing poor polymorphisms were regarded as candidate markers. The core SNPs were used to obtain genotypic data from the sugarcane germplasm accessions, and the online software, Caoliaoerweima (http://cli.im/, accessed on 17 May 2022), was used to generate the QR codes for sugarcane. The genotypes based on the SNP barcodes were entered, and the QR codes were automatically generated. When the barcode is scanned, the genotype of each accession is shown.

## 3. Results

### 3.1. Sequencing Data Analysis and SNP Identification

A total of 103 libraries for sugarcane genomic DNA samples were constructed after quality control and data filtering. These libraries yielded 223.63 Mb raw Illumina reads by SLAF sequencing. The GC content of the 103 samples ranged from 42.08% to 45.69%, with an average of 44.13%. The quality values of the sequenced bases Q30 ranged from 89.41% to 93.08%, with an average of 90.91%. A total of 243,526 SLAF tags were developed with an average sequencing depth of 11.04 x. The average mapping ratio was 97.89% (Appendix A). The average comparison rate met the requirements of the sequence analysis such that subsequent analyses could then be performed.

After completing the sequencing, SNP site variant identification on the tested germplasm was carried out using the Genome Analysis Toolkit (GATK; https://gatk.broadinstitute.org/hc/en-us, accessed on 15 June 2019) process and samtools. The identified SNPs were filtered, and 105,325 high-quality SNPs were obtained for subsequent analyses. An analysis of the types of predicted mutations associated with these SNPs showed that of the 6 possible single-base mutations, C/T and A/G transitions were the most frequent, accounting for 39.59% and 39.51% of the total, respectively. Of the 105,325 SNPs, 83,317 were transitions and 22,008 were transversions (Figure 1A). Further analysis of the SNP distribution in the genome found that 83.96% were located in intergenic regions, 4.16% were located in introns, 1.30% were located in exons, 5.49% were located in the 5-kb region upstream of the transcriptional start site, 4.82% were located in the 5-kb region downstream of the transcription termination site, 0.06% and 0.12% were located in the 5′ and 3′ UTRs, respectively, and 0.06% were located in the splice junctions (Figure 1B). Functional annotation of the SNPs in the exonic regions of the genes is shown graphically in Figure 1C. There were 4427 non-synonymous single-nucleotide variants (SNVs) that are predicted to cause changes in the encoded amino acid, and 2514 synonymous SNVs that are predicted not to change the amino acid sequence. The ratio of non-synonymous SNVs to synonymous SNVs is 1.76. In addition, 182 SNPs are predicted to produce a premature stop codon (stop gain) and 103 SNPs were predicted to abolish a stop codon (stop loss).

### 3.2. Genetic Variations and Population Structure

PCA showed that the first and second axis captured 35% and 3.1% of the overall variance, respectively. The 103 sugarcane accessions were divided into two well-separated clusters and combined with wild stem sugarcane and modern sugarcane accessions (Figure 2A). Although there were large differences between wild and modern sugarcane accessions, modern sugarcane accessions from the same geographical area are irrelevant. We also used the neighbor-joining method to construct a phylogenetic tree for the 103 sugarcane accessions. The 24 wild sugarcane accessions are in the analysis cluster together and are distinguishable from the 79 sugarcane cultivars in the study (Figure 2B). The 79 sugarcane accessions combined with some subgroups do not fully cluster by geographical origin, which is consistent with the PCA results. The genetic structures of the 103 sugarcane accessions were analyzed with different clusters (*K* from 1 to 10) using the cross-validation error rate. The cross-validation error-validation error rate was lowest when *K* = 2, which shows that the 103 sugarcane accessions can indeed be grouped into two clusters (Figure 2C,D). Moreover, all the test materials were completely classified, which is consistent with the phylogenetic tree and PCA.

### 3.3. SNP Primers

A total of 192 SNP loci were selected to be transformed into PCR-based markers. A 200-bp sequence up-and-downstream of the SNP sites was used to fill sites with high homology sequences through BLAST analysis in NCBI, after which 72 SNP sites remained. Amplification primers and extension primers were designed for these loci and based on the experimental situation where 43 qualified SNP markers remained (Figure 3). Polymorphic sites among the 180 sugarcane materials were considered to be core SNP markers and non-polymorphic sites were considered as candidate SNP markers. There were 32 and 11 core and candidate SNP markers, respectively (Appendix A).

### 3.4. Construction of DNA Fingerprints

The overall predictive accuracy of the 32-SNP barcode was 100% for the 180 sugarcane germplasms. The barcode can distinguish all materials obviously and effectively (Figure 4). The online software, Caoliaoerweima (http://cli.im/, accessed on 1 June 2022), was used to encode the genotypic data for the core SNPs in the 180 sugarcane accessions. A QR code fingerprint was constructed for each sugarcane line. The fingerprints contained corresponding information, such as variable name, type, and botanical classification. QR codes that are easily scanned and obtained using a mobile internet device, such as a cellphone, were translated from the fingerprints for these accessions. For example, the QR code for FuNong 41 shown in Appendix A contains the following information: Name: Funong41, Cultivars, Monocotyledons, Poales, Gramineae, Saccharum L, Characteristics: belong to ripe, high sugar, high yield varieties. Plant tall, large stem, plant growth erect, stem slightly front; Internode round tubular, with short shallow bud groove; Sugarcane stem uniform, shading part of red, exposed part of purple; No growth crack and cork plaque, wax powder thick, no air rooting; Bud oval shape, bud base from leaf mark, bud tip but growth band, growth is not prominent, with light yellow to yellow, purple after exposure; Root points in 2–3 rows, irregularly arranged; Yellow-green leaf sheath, older leaf sheath with red strips, loose stem, easy to defoliate, no. 57 hair group is not developed; Leaves dark green, long, medium wide, oblique, about 1/3 of the bent; The inner auricle is triangular and the outer auricle is missing. Parental source: ROC20 × Yuetang91-976. Breeding institutions: National Engineering Research Center of Sugarcane, Fuzhou, China. Fingerprint code: G/A, C/T, C/A, N/N, G/G, C/T, A/A, N/N, T/T, G/A, C/C, T/T, T/T, G/G, G/A, G/G, G/G, T/T, T/A, T/T, C/T, G/G, C/C, C/T, C/T, T/T, T/T, C/T, C/T, C/C, A/A, G/G. In addition, there are pictures of the accession, cane buds, and stems.

### 3.5. Ability of Core SNPs to Identify Sugarcane Accessions

Using the genotyping data of the 32 core SNP markers, a genetic distance matrix for the 180 sugarcane accessions was calculated. The genetic distance between sugarcane accessions ranged from 0 to 0.743, and the mean genetic distance was 0.133. The value for the genetic distances was mainly distributed in the range of 0–0.1 and 0.1–0.2 (Appendix A), indicating that the genetic distance among most sugarcane materials was close, although for a few sugarcane materials the distance was large. The 180 sugarcane materials are distributed on each branch of the cluster analysis diagram, showing that the 180 accessions could be clearly identified by the 32 core SNPs (Appendix A).

## 4. Discussion

### 4.1. SNPs Show High Efficiency for Genotyping

SSR markers have been widely utilized in plant genomic mapping, selection identification, association analysis, and marker-assisted breeding over the past three decades. SSR markers may undergo mutations along the length of the SSR loci during PCR amplification or as a result of DNA enzyme slippage and duplication, which can lead to enormous numbers of shutter bands during polymorphism detection that make it difficult to interpret bands and construct multiple SSR detection systems. In contrast, SNPs have a low mutation rate and might be applicable in high-throughput automatic detection. The establishment of a SNP complicated typewriting system exploitation multiplex PCR and exposure typewriting strategies is comparatively easy. In this study, 7869,726 SNP loci were found to facilitate SLAF sequencing, and 32 high-quality SNP markers were identified to create sugarcane fingerprints. When compared to mass spectrometry or digital PCR strategies, this application can simultaneously examine several SNP sites within the same reaction tube. In addition, the capillary ionophoresis analysis of microsequencing merchandise using SNaPshot is quite practical, easy, and straightforward.

### 4.2. Advantages for SLAF-Sequence and Multiplex SNaPshot Panels Systems

The application of the highly-developed and popular SLAF-sequence simplified genome sequencing approach has been extensively documented for a variety of crops, including Chinese lou onions [52], sweet potatoes [53], broomcorn millets [54], and others. It is more cost-effective than re-sequencing and can also produce a large number of SNPs with a uniform distribution. The “small sequencing” or “SNaPshot” typing technology is based on the fluorescence-labeling single base extension approach. Less is known about medical investigations of human disorders, but similar studies of tea and Arabidopsis have also been described. In our work, this technique is capable of running 20 PCR reactions concurrently in one machine. Genemapper 4.1 [51] can determine the base mutations of SNP sites in the same location of various sugarcane materials based on the hue of various fluorophore groups. Using KASP technology for the genetic analysis of the same population, numerous crops, including maize [55], wheat [56], cabbage [57], and others, have been assessed before we started our research. The SNaPshot technology is slightly more expensive than the KASP technology, but it also performs better, has a higher throughput, and is more site-adaptable [58,59]. We selected the technological approach most suited for our work in light of the complexity of the sugarcane genome, which is an allopolyploid crop.

The advantages and downsides of the aforementioned two strategies are weighed against our level of economic tolerance and the sugarcane genomic characteristics to arrive at the best possible combination. The tetraploid genome used in our experimental study only comprises 32 chromosomes. The cultivated species, on the other hand, are a product of multi-generational selection and are derived from tropical species. The genetic data on chromosomes is extensive and intricate. The cultivated species, nonetheless, were chosen over multiple generations and descend from tropical species. There is a wealth of complex genetic data on the chromosomes. At the moment, the genome data for tropical species have been made public, and the genome assembly data for other materials will soon follow. Additionally, this technology system has a great deal of room for improvement. We may be capable of directly obtaining specific SNP loci associated with genes as heavy sequencing technologies reduce costs as well as the initiation of other tag classification methods, and even some functional gene sequences of related reports. This will help to use fewer molecular marker materials to determine similarities and differences. Moreover, the authenticity and efficiency will be improved compared to the current level.

### 4.3. SNP-Based Genetic Relationships among Sugarcane Accessions

Recently, the genus for sugarcane and its relatives was preserved and wont to guide sugarcane breeding. The hybrid varieties, hybrid oldsters, and intermediate materials obtained through genetic improvement are various. These accessions are evaluated for victimization in completely different classes, like landraces, historical and up-to-date cultivars, breeding materials, and wild germplasms. To facilitate subsequent breeding and application of germplasms, an associated understanding of the genetic relationships, and therefore the structure of population biology among varieties/accessions at the genomic level, is required.

Using the nucleotide sequences of the variant site SNPs obtained by SLAF for the 103 sugarcane accessions in this study, the results for PCA, genetic population analysis, and phylogenetic analysis were consistent. The clustering results showed that the wild germplasm fell into a single cluster, which also reflected the substantial genetic differences between wild germplasms and cultivars. However, cultivated accessions were distributed in different groups, indicating that these cultivars have no obvious correlation according to geographical region. The genetic background of the population is complex and diverse. *S. spontaneum* L., a wild material, is gathered in a single cluster, whereas cultivar materials are distributed among different communities. In the long-term domestic and breeding process, the genetic background of different wild accessions is introduced to the genome of cultivated species [60]. These sugarcane germplasm exchanges led to an indistinct correlation between the results of population genetic analysis and geographical origin. Our results provide a favorable reference and basis for the better utilization of sugarcane germplasm resources and could be a valuable guide for the selection of parental lines in future sugarcane breeding programs.

### 4.4. DNA Fingerprinting Based on SNP Markers

DNA fingerprints, which were developed in 1986 by Jefferys, have the advantage of being fast and accurate, and are based on DNA sequence polymorphisms at multiple loci distributed across the genome. The fingerprint map for sugarcane accessions is rich in polymorphisms and has a high degree of individual specificity and environmental stability that allows different individuals to be discriminated. DNA fingerprinting is a powerful tool to identify varieties and is also suitable for identifying plant germplasm resources. RFLP, AFLP, SSR, and SNP have been successively used to construct fingerprints. SSR fingerprinting is often used for variety identification and analysis due to its good repeatability, simplicity, ease of operation, and mostly co-dominant markers. Pan et al. developed the SSR marker-based molecular identification database for sugarcane [61], which provides a molecular description for new variety registration articles and enables sugarcane breeders to identify mislabeled sugarcane clones in crossing programs and determine the paternity of cross progeny [62]. Ali used 21 pairs of sugarcane genome SSR primers to establish the molecular fingerprint database of 91 sugarcane varieties [63]. The results of that study provide a reference for establishing a SSR standard fingerprint database. SSR markers also reveal several practical flaws, such as a restricted number of markers, few detection sites, a particular rate of locus mutation, and sensitivity to mutation response. Based on these factors, SNPs are the preferred marker for building a DNA fingerprint database. In sugarcane, however, no SNP marker-based fingerprints were available. Thus, creating an SNP marker-based fingerprint is critical for sugarcane variety specificity and authenticity verification, as well as genetic relationships. The results obtained from SNP-based fingerprints will enrich the availability of genome information for sugarcane and could be used for further study of genetic diversity and modern molecular breeding.

SNP-based high-throughput assay techniques and data analysis are based on the genetic characteristics of “binary variants”, in which two different fluorophores are used to label two different alleles and fluorescence detection systems that can effectively distinguish between two homozygous genotypes and one heterozygous genotype. To meet the needs of large-scale detection and fingerprint database construction, multiple SNP markers must be developed and screened to achieve the ideal ability to identify different varieties. In wheat, SNP markers were compared with other DNA markers to identify SNPs that have the necessary characteristics for high-throughput assays that have high accuracy [64]. In tomato (*Solanum lycopersicum* L.), a custom-made Illumina SNP-panel was used to genotype 214 accessions and examine polymorphism patterns [65]. This study reported SNPs characterized by high accuracy and a simple operation that provides a new approach for future research and development of genetic resources for tomato crops.

The complexity of species genomes, marker types, and marker detection techniques, as well as varying numbers, influence the choice of key fingerprint-identifying markers. In the present study, SNaPshot technology was used to detect SNPs, and 32 core SNP loci were used to create fingerprints for 180 sugarcane germplasm resources, such that each variety had its unique fingerprint code. Markers for SNaPshot are highly polymorphic and have an enhanced capacity to identify different germplasms, thus providing a powerful resource for genotyping and identifying sugarcane cultivars. SNaPshot also represents a useful method for classifying sugarcane germplasm and ensuring the authenticity and purity of varieties. The QR codes provide convenient access to information on which decisions for sugarcane breeding programs can be made.

### 4.5. Application for Fingerprinting and Identification of SNPs

Identification and screening of new varieties or cultivars are important in crop breeding. Traditionally, new varieties are identified using morphological characteristics. However, this approach has several disadvantages, including long identification cycles, high costs, and the confounding effects of environmental factors. Moreover, the genetic background of modern sugarcane cultivars became narrower since only a few elite parental lines have been used. In turn, the existing morphological genetic variation decreased and was more limited in terms of variety identification. Therefore, the use of DNA molecular markers to identify germplasm on a molecular level is critical. Through SLAF sequencing, we obtained 192 SNPs using filtering and screening, and of these 32 core markers, 11 candidate markers were successfully developed. Fingerprints for 180 sugarcane accessions were successfully constructed using 32 core markers.

Variety identification should be accurate, efficient, simple, and have a high throughput. The detection method should be suitable for automation, which is a requirement for a variety of identification technologies in the future. SNP markers can be used to identify existing sugarcane resources, laying a foundation for the standardization of sugarcane and its application in subsequent molecular genetics and breeding programs. At present, high-throughput detection of SNP markers in large sample sets has shown advantages for the rapid identification of different varieties. In the future, high-throughput SNP detection technology will have broad applications for the fingerprinting and identification of newly-developed sugarcane varieties.

## 5. Conclusions

In this study, SLAF-seq was used to analyze the population structure and genetic diversity of 103 sugarcane germplasms. The population could be divided into two well-separated clusters with no significant correlation between the distribution of sugarcane germplasm and its geographical origin. SNaPshot multiplex technology was used to develop 43 SNP markers (32 core SNPs and 11 candidate SNPs) for germplasm identification and fingerprint construction. Precision, high flux, and excellent site adaptation are benefits of the SLAF-sequence and Multiplex SNaPshot Panels system. Given the complexity of the sugarcane genome, the research goal of creating fingerprints, and the need to identify sugarcane germplasm resources, this technique is highly innovative. The fingerprints of 180 sugarcane germplasm could be constructed based on the 32 core SNPs. The results of this study provide a useful approach for genotyping, classifying, and identifying germplasm and resources in sugarcane breeding.

## Figures and Tables

**Figure 1 genes-13-01477-f001:**
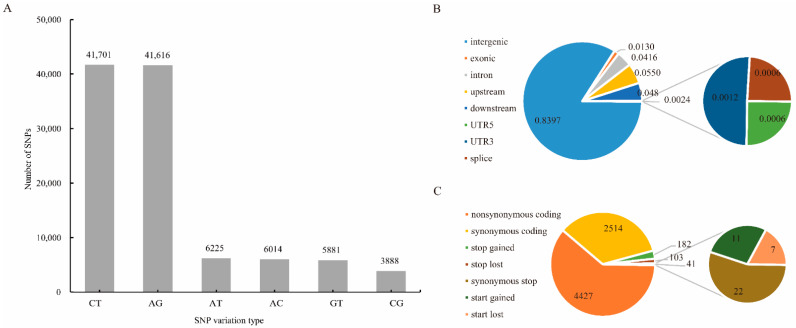
Analysis of SNPs in sugarcane germplasm accessions. (**A**) The six SNP types and the number of SNPs of each type. (**B**) The positions of the SNPs in the gene structures. Upstream: the SNP is located in the region 5-kb upstream (5′) of a gene; Downstream: the SNP is located in the region 5-kb downstream (3′) of a gene; Splicing: variable splicing site within 2 bp. (**C**) Annotations of the SNPs in the exons. Non-synonymous coding: a single-nucleotide change that causes an amino acid change; Synonymous coding: a single-nucleotide change that does not cause an amino acid change; Stop gain: the mutation causes early termination of translation; Stop loss: the variation causes the loss of the terminator codon.

**Figure 2 genes-13-01477-f002:**
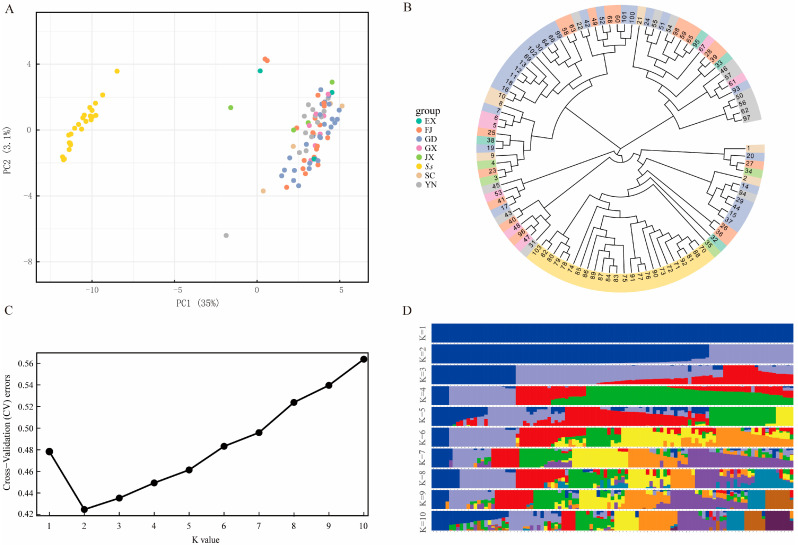
Population genetic structure analysis of 103 sugarcane germplasms based on polymorphic SNP sites. (**A**) Principal component analysis (PCA). (**B**) An unrooted neighbor-joining (NJ) phylogenetic tree. (**C**) Cross-validation error rates corresponding to different *K* values. (**D**) Population structure of the 103 sugarcane germplasm resources.

**Figure 3 genes-13-01477-f003:**
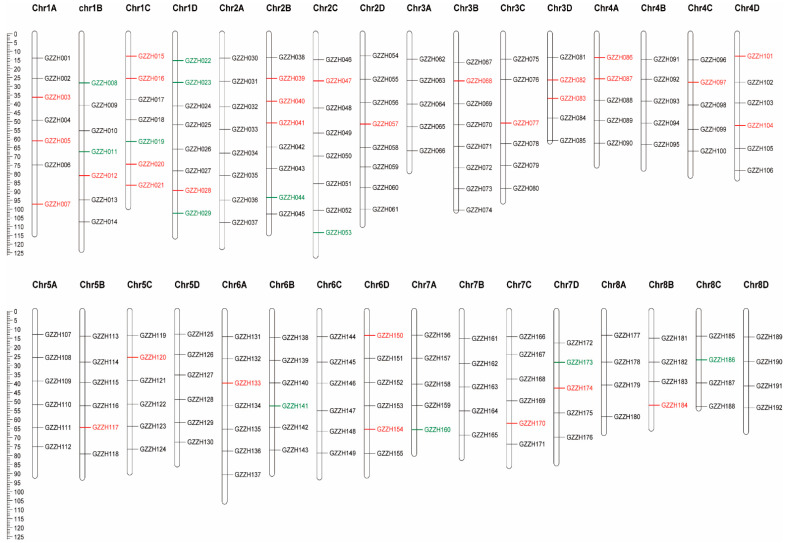
Distribution of SNP markers on sugarcane chromosomes. Red represents the core SNP markers, green represents the candidate SNP markers.

**Figure 4 genes-13-01477-f004:**
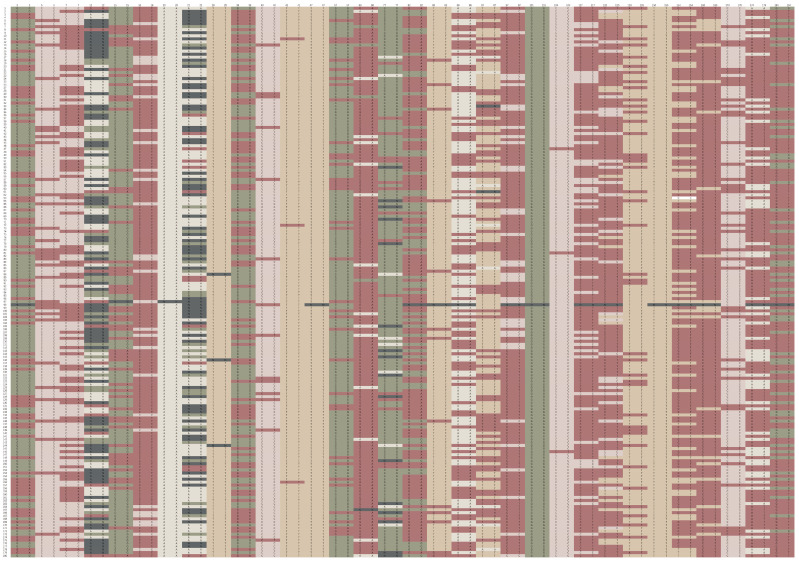
Fingerprints of 180 sugarcane accessions Each line represents one SNP locus and each column represents one accession. Green (BCCBB0), apricot (D7CAB1), caesious (9C9EB9), light pink (DCCFCB), and plain grey (525B66) represent A/A, T/T, G/G, C/C, and N/N, respectively. Heterozygous genotypes are shown in claret-red (A77979).

## Data Availability

Not applicable.

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
