# Peer review of "Development of SLAF-Sequence and Multiplex SNaPshot Panels for Population Genetic Diversity Analysis and Construction of DNA Fingerprints for Sugarcane"

_genes, 2022, doi:10.3390/genes13081477_

Round 1

Reviewer 1 Report

Many sentences do not have the correct punctuation and it is difficult to read the text.

English should be improved; grammar need for enhancement in many sentences and paragraphs.

All figures need for resolution enhancement.

Figures is not in printable quality. Also, some portions of the texts are losing their readability while sizing the image as per text area.  Kindly provide a better quality figure.

Please check the References in-text and end-list for uniformity in style.

The conclusion you have provided is quite brief and provides sufficient feedback on the main objectives of your study.

The list of REF. is repeated twice. Please delete one 

Reviewer 2 Report

A review of Development of SLAF-sequence and Multiplex SNaPshot panels for Population Genetic Diversity Analysis and Construction of DNA Fingerprints for Sugarcane: A few points deserve attention for further publication. I suggest that it is accepted for publication after the following revisions:

- The authors could clarify in the manuscript's abstract the mechanism, advantages, problems, and solutions for the Development of SLAF-sequence and Multiplex SNaPshot panels.

- In addition, the authors should highlight the advantages/disadvantages of this Development of SLAF-sequence and Multiplex SNaPshot panels methods for industrial application and how this information will be addressed in the manuscript.

- Advantages for Development of SLAF-sequence and Multiplex SNaPshot panels systems: Which methods have advantages? Are they simple methods of contribution? When compared with other sustainable techniques? Authors must leave this clear information throughout the text and the methods discussed in this manuscript. In addition, this information is needed for the Development of SLAF-sequence and Multiplex SNaPshot panels systems contribution protocols are applied on an industrial scale.

- Problems with Development of SLAF-sequence and Multiplex SNaPshot panels systems: Does this protocol have a significant problem? This discussion could be improved.

- Additionally, advances in the Development of SLAF-sequence and Multiplex SNaPshot panels  systems with engineered tailor-made have been performed with other strategies. May open new opportunities. This discussion could be improved.

- This manuscript has broached interest in the progress and recent applications of Development of SLAF-sequence and Multiplex SNaPshot panels: The main contributions to the accomplishment of this work must be included in the conclusion.

 - Please check all references according to the author's instructions.

- The manuscript must be formatted according to the journal's standards.
